# SEL-BALD: Deep Bayesian Active Learning with Selective Labels

**Ruijiang Gao**
Naveen Jindal School of Management
University of Texas at Dallas
Richardson, TX 75082
`ruijiang.gao@utdallas.edu`

**Mingzhang Yin**
Warrington College of Business
University of Florida
Gainesville, FL 32611
`mingzhang.yin@warrington.ufl.edu`

**Maytal Saar-Tsechansky**
Information, Risk, and Operations Management
University of Texas at Austin
Austin, TX 78712
`maytal@mail.utexas.edu`

## Abstract

Machine learning systems are widely used in many high-stakes contexts in which experimental designs for assigning treatments are infeasible. When evaluating decisions is costly, such as investigating fraud cases, or evaluating biopsy decisions, a sample-efficient strategy is needed. However, while existing active learning methods assume humans will always label the instances selected by the machine learning model, in many critical applications, humans may decline to label instances selected by the machine learning model due to reasons such as regulation constraint, domain knowledge, or algorithmic aversion, thus not sample efficient. In this paper, we study the Active Learning with Instance Rejection (ALIR) problem, which considers the human discretion behavior for high-stakes decision making problems. We propose new active learning algorithms under deep bayesian active learning for selective labeling (SEL-BALD) to address the ALIR problem. Our algorithms consider how to acquire information for both the machine learning model and the human discretion model. We conduct experiments on both synthetic and real-world datasets to demonstrate the effectiveness of our proposed algorithms.

## 1 Introduction

Machine learning is increasingly deployed in high-risk applications, including medical diagnosis, fraud detection, and criminal justice [Zeng et al., 2017, Cecchini et al., 2010, Savage, 2020]. Given enough labeled data and the same set of feature input, a machine learning model may achieve better performance than human experts [Abramoff et al., 2023]. Therefore, there have been growing efforts to develop machine learning (ML) models to assist humans in improving high-risk decisions.

While modern deep learning algorithms' strong performance across applications makes them particularly promising for supporting experts, they are often data-hungry and require a large amount of labeled data to perform well. Yet, in many impactful high-risk applications acquiring a large number of labeled data is highly costly or can be prohibitive, such as when labeling necessitates costly human experts' time and effort. For instance, labeling insurance claims requires human experts' resources to establish if an insurance claim is fraudulent; similarly, determining medical diagnoses may require diagnostic procedures that may be time-consuming and possibly risky for patients. Active learning methods can be a promising path to address this problem, given they aim to select especially

38th Conference on Neural Information Processing Systems (NeurIPS 2024).

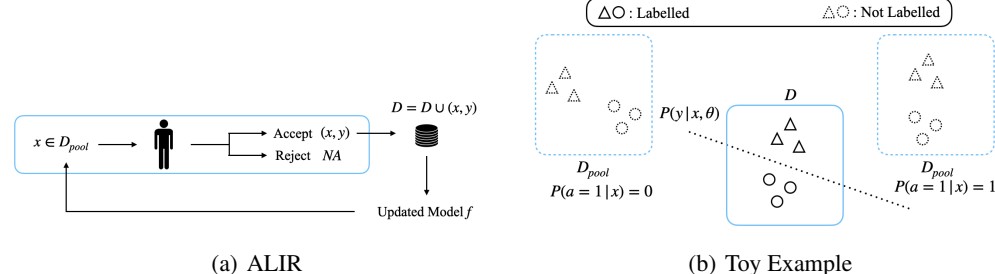

(a) ALIR               (b) Toy Example

Figure 1: (a): Active Learning with Instance Rejection (ALIR). The requester wants to acquire additional labeled samples to improve the ML classifier $f$. Due to the selective labels problem, only accepted labeling requests are labeled and added to the dataset $\mathcal{D}$. (b): The most informative samples to the current ML classifier may not be the best samples to label in ALIR. The instances on the left of $\mathcal{D}_{\text{pool}}$ are more informative to the current classifier yet humans will never label these instances.

informative instances for labeling to minimize the number of labeled instances necessary to achieve desirable performance [Cohn et al., 1996].

To do so, most active learning methods operate iteratively, where an initial machine learning model is first trained on a small labeled data set and the model then informs the selection of particularly informative instances to be labeled. A new model is then retrained on the augmented labeled data and this process is repeated until a desirable goal is achieved.

However, most active learning methods assume instances selected for labeling by the algorithm will always be labeled [Settles, 2009]. However, in many real-world high-risk contexts, such as acquiring diagnoses through costly invasive diagnostic procedures or obtaining the conclusions of insurance fraud investigations, experimental design is infeasible and it is not possible to automatically acquire any label the active learning algorithm deems informative. Rather, requests for label acquisition must be considered and approved by a human expert. Indeed, some diagnostic biopsies may *never* be approved by the human doctor because they are inconsistent with medical practices [Dai and Singh, 2021]; if the requests are declined, the biopsy outcomes will not be revealed to benefit the machine learning model. Similarly, an insurance fraud agent may decline a request for an investigation because of the remote location or because of a prior belief that the case is unlikely to be fraudulent. Any given human discretion behavior over requests to acquire labels at a cost is *unknown* and can be influenced by various factors. This is known as the *selective labels* problem [Lakkaraju et al., 2017, Kleinberg et al., 2018, De-Arteaga et al., 2018, Wei, 2021], in which the outcomes are more likely to be available for a subset of instances. When a human expert must approve the decision to obtain a label, selective labeling would result from the expert's discretion over the labeling process.

Independently of the labeling costs, human decision makers also need to invest time and effort in considering the algorithm's recommendation before making a decision. For example, the insurance agent needs to evaluate each case in order to decide whether an investigation should be launched and doctors will go through patients' medical record and assess the risk prior to approving a diagnostic procedure. Therefore, our goal is to maximize the machine learning model's predictive performance within a total budget considering both the examination and labeling cost.

Ultimately, while traditional active learning focuses on identifying instances that would be most informative for learning, when the human must approve each labeling request, the human may be inundated with requests to consider acquisitions, any portion of which may be declined and thus not benefit learning, while incurring a significant loss of the human expert's resources. This poses a new challenge for active learning in which it would be valuable to bring to bear the human discretion of any labeling request so as to identify labeling requests that can achieve the most improvement in model performance for a given labeling and human discretion costs. We refer to this problem as Active Learning with Instance Rejection (ALIR), illustrated in Figure 1(a). Crucially, in ALIR, the instances deemed most informative for the current ML classifier may not be the best candidates for labeling, as humans may choose to reject these instances (as shown in Figure 1(b)). This challenge highlights the need for developing new active learning methods tailored to the ALIR framework.

In this paper, we make the following contributions:

- We study the Active Learning with Instance Rejection (ALIR) problem that considers the human discretion behavior in high-stakes selective labeling contexts. Unlike conventional active learning, where every instance is labeled by humans, ALIR allows humans to exercise discretion, selectively choosing whether or not to label certain instances.

- We propose different active learning algorithms for ALIR under the bayesian active learning with disagreement framework and discuss their trade-offs. Unlike traditional active learning, the benefit of each method depends on the underlying human discretion behavior. We show theoretically that ALIR is equivalent to traditional active learning when the human discretion behavior is homogeneous across instances and propose new methods for unknown, heterogeneous discretion behaviors.

- We conduct comprehensive experiments on both synthetic and real-world datasets to demonstrate the effectiveness of our proposed algorithms.

## 2 Related Work

**Active Learning:** To render costly data annotation more efficient, Active Learning has been studied extensively in the machine learning community [Lewis and Gale, 1994, Schohn and Cohn, 2000, Nguyen and Smeulders, 2004, Freytag et al., 2014, Gal et al., 2017, Sener and Savarese, 2017, Gong et al., 2019, Gao and Saar-Tsechansky, 2020]. Traditionally, the key challenge addressed by most methods is to label the most informative unlabeled instances from which to induce the classifier. Existing methods assume humans will always label the instances selected by the active learning algorithm. We consider the selective labels problem where in many high-stakes decision making problems, humans may not label the selected instances due to their past experience, domain knowledge, or algorithmic aversion[Dai and Singh, 2021, De-Arteaga et al., 2018, Dietvorst et al., 2015]. Recent papers have studied active learning for treatment effect estimation from observational data [Jesson et al., 2021, Deng et al., 2011, Sundin et al., 2019]. Unlike these papers, we study the selective labels setup where the outcome is selectively observed, which is different from the treatment effect estimation problem where we can observe an outcome under the selected treatment arm. The ALIR problem is also known as the Active Learning with Abstention Feedback problem and studied by other papers [Fang et al., 2012, Yan et al., 2015, Amin et al., 2021, Nguyen et al., 2022, Robinson et al., 2024]. Fang et al. [2012] is close to the proposed $e$-BALD method but we use the bayesian neural network for modeling human behaviors. Yan et al. [2015] additionally models noisy human feedbacks. Amin et al. [2021] considers a different setting where a machine can determine whether human or the AI should label the instance. A concurrent work [Robinson et al., 2024] motivates the problem from the e-commerce perspective where users may not click on the ads the platform sent and not observing the purchase label. Unlike their work, we motivate the problem from the human-AI perspective and provide information-theoretical results for the proposed criteria with a focus on the deep bayesian active learning setting.

**Selective Labels:** Lakkaraju et al. [2017] proposes the "contraction" technique to evaluate a machine learning classifier's performance in the selective labels problem. However, the point identification of the model performance is generally impossible without strong assumptions and partial identification is studied in the recent literature [De-Arteaga et al., 2018, D'Amour, 2019, Guo et al., 2022, Yin et al., 2024]. Unlike previous papers on studying the limitations in evaluation and optimizing predictive rules in the selective labels problem, we focus on how to acquire useful information to improve the machine learning model in the selective labels problem. Rambachan and Roth [2019] studies how historical human-generated data may bias the machine learning model in the selective labels problem and our paper studies how to work with the human decision makers to improve the machine learning model in selecting informative samples.

**Human-AI Collaboration:** There are works in the machine learning community designing different forms of human-AI collaborated decision-making systems such as learning to defer [Cortes et al., 2016, Madras et al., 2018, Gao et al., 2021, Gao and Yin, 2023, Gao et al., 2023], or learning to advise [Bastani et al., 2021, Grand-Clément and Pauphilet, 2024, Bhatt et al., 2023, Cao et al., 2024]. Unlike these work, we focus on how to collect information under human supervision. Our work is also motivated by the literature on overriding decisions and trust in algorithm adoptions [Sun et al., 2022, Wang et al., 2022]. Dietvorst et al. [2015] finds that humans are likely to override the

algorithm's decisions when they observe machine learning algorithms make mistakes. Bansal et al. [2021] studies how the quality of the explanation and the predictive quality of the machine learning model affect human decisions. There are also works that study different interventions to improve the trust of humans in the machine learning model [Dietvorst et al., 2018, Lakkaraju and Bastani, 2020, Zhang et al., 2020]. de Véricourt and Gurkan [2023] uses an analytical modeling framework and stylized human behavior model to show that humans may never know the true performance of AI in the selective labels problem. Our work is different from these papers as we focus on how to acquire useful information to improve the machine learning model in the selective labels problem.

## 3 Problem Statement and Background

### 3.1 Selective Labels Problem

We have access to a pool of unlabeled data $\mathcal{D}_{\text{pool}} = \{x_i\}_{i=1}^n$, where each instance has a underlying label $y_i \in \{1, \cdots, K\}$. We denote the marginal distribution of $X$ as $P(X)$ and the conditional distribution of $Y$ given $X$ as $P(Y|X)$. We assume that the unlabeled data is drawn i.i.d. from $P(X)$. A human decision maker will make a yes decision ($a_i = 1$) or no decision ($a_i = 0$) for the decision subject. The corresponding outcome $y_i$ can only be observed when $a_i = 1$.

We assume there exists a human discretion function $e(X)$ that maps the input $X$ to a probability of being labeled by the human decision maker. We denote the probability of $X$ being labeled as $P(A = 1|X) = e(X)$. This corresponds to how humans will examine and label the instances. Then for an instance $x_i$, if $a_i = 1$, it will be labeled, returning the label $y_i$, and be added to the current labeled dataset $\mathcal{D} = \{x_i, y_i\}_{i:a_i=1}$. Otherwise, the human labeler will reject it and no outcome will be available for the instance. No matter whether the instance is labeled or not, $(x_i, a_i)$ will be added to the current human decision dataset $\mathcal{L} = \{x_i, a_i\}_{i=1}^n$, which records all the human labeling information. A predictive model with likelihood $P(y|x, \theta)$ parametrized by $\theta \sim P(\Theta|\mathcal{D})$ can then be trained on the labeled dataset $\mathcal{D}$.

### 3.2 Active Learning

The requester now wants to acquire some additional labeled samples to improve the ML classifier. Since human labelers need time and resources to decide whether to investigate each instance (e.g., have a meeting to decide whether to operate on the organ, decide on whether to audit the firm) and label it (e.g., perform the biopsy, audit the firm), the requester wants to minimize the number of instances that need to be labeled. We denote the cost for examining the instance as $c_e$ and the cost for labeling the instance as $c_l$. The total cost is the sum of the current examination cost and label cost. Therefore, with a total budget of $B$, the requester wants to select a subset of instances $\mathcal{S} \subseteq \mathcal{D}_{\text{pool}}$ to be labeled by the human labelers while maximizing the performance of the ML classifier $P(y|x, \theta)$.

Given a model $P(y|x, \theta)$, pool dataset $\mathcal{D}_{\text{pool}}$, and labeled dataset $\mathcal{D}$, active learning algorithms use an acquisition function $\mathcal{A}(x, \theta)$ to select where to label next by $\arg\max_{x \in \mathcal{D}_{\text{pool}}} \mathcal{A}(x, \theta)$. For example, $\mathcal{A}(x, \theta) = \text{unif}()$ corresponds to the random acquisition function, which $\text{unif}()$ returns a uniformly random number between 0 and 1; and $\mathcal{A}(x, \theta) = \mathbb{H}(y|x, \mathcal{D}) = \sum_c P(y = c|x, \mathcal{D}) \log(P(y = c|x, \mathcal{D}))$ corresponds to the uncertainty sampling principle that picks the next sample with the largest predictive entropy. Our algorithm extends the following Bayesian Active Learning with Disagreement (BALD) algorithm [Houlsby et al., 2011, Gal et al., 2017].

**Bayesian Active Learning with Disagreement (BALD):** BALD [Houlsby et al., 2011] uses the epistemic uncertainty as the aquisition function. The epistemic uncertainty measures the uncertainty about model parameters instead of the inherent uncertainty about the data that cannot be reduced by acquiring more samples. The information gain is defined as the mutual information between the label and the model parameters which can be written as follows:

$$\mathbb{I}(y, \theta|x, \mathcal{D}) = \mathbb{E}_{y|x, \mathcal{D}}[\mathbb{H}(\theta|\mathcal{D}) - \mathbb{H}(\theta|y, x, \mathcal{D})]$$

$$= -\sum_c P(y = c|x, \mathcal{D}) \log(P(y = c|x, \mathcal{D})) + \mathbb{E}_{\theta \sim P(\theta|\mathcal{D})}\left[\sum_c P(y = c|x, \theta) \log(P(y = c|x, \theta))\right].$$

(1)

Here the posterior $P(\theta|\mathcal{D})$ can be approximated by the approximated posterior $q(\theta|\mathcal{D})$ from the Bayesian neural network, and using MC sampling [Gal et al., 2017] to approximate Equation (1) as

$$\approx -\sum_c \left(\frac{1}{T}\sum_t \hat{p}_c^t\right)\log\left(\frac{1}{T}\sum_t \hat{p}_c^t\right) + \frac{1}{T}\sum_{c,t}\hat{p}_c^t\log(\hat{p}_c^t),$$

where $\hat{p}_c^t$ is the probability of input $x$ taken class $c$ under the $t$-th MC sample from the approximated posterior $q(\theta)$. We denote this objective as $\hat{\mathbb{I}}(y,\theta|x,\mathcal{D}) \approx \mathbb{I}(y,\theta|x,\mathcal{D})$.

BALD will then select the instance by $\arg\max_{x_i\in\mathcal{D}_{\text{pool}}}\hat{\mathbb{I}}(y,\theta|x_i,\mathcal{D})$.

## 4 Methods

### 4.1 Naively Apply Bayesian Active Learning with Disagreement (BALD)

If we ignore the selective labels problem and simply treat it as the traditional active learning setup where every instance will always be labeled, we can use the same computation procedure in the standard BALD algorithm (in the supervised learning setup) to select the next instances to label.

**Definition 4.1** (Naive-BALD)**.**

$$\hat{\mathbb{I}}_{\text{Naive}}(y,\theta|x,\mathcal{D}) := \hat{\mathbb{I}}(y,\theta|x,\mathcal{D},a=1)$$
$$= -\mathbb{E}_{\theta\sim P(\theta|\mathcal{D})}\sum_c P(y=c|x,\theta,a=1)\log\left(\frac{P(y=c|x,\mathcal{D},a=1)}{P(y=c|x,\theta,a=1)}\right). \quad (2)$$

However, when humans are involved, we need to consider the probability of each instance being labeled. When the instance with high information gain is unlikely to be labeled, BALD may send many instances to the human labelers but only few of them will be labeled, thus wasting the time and resources of the human labelers.

### 4.2 Bayesian Active Learning with Disagreement with Selective Labels (SEL-BALD)

First, we show the information theoretical objective under ALIR. When considering the human discretion behavior, the information gain $\hat{\mathbb{I}}(y,\theta|x,\mathcal{D})$ can be derived as

**Definition 4.2** ($e$-BALD)**.**

$$\hat{\mathbb{I}}(y,\theta|x,\mathcal{D}) = e(x)\hat{\mathbb{I}}(y,\theta|x,\mathcal{D},a=1)$$
$$= -e(x)\mathbb{E}_{\theta\sim P(\theta|\mathcal{D})}\sum_c P(y=c|x,\theta,a=1)\log\left(\frac{P(y=c|x,\mathcal{D},a=1)}{P(y=c|x,\theta,a=1)}\right) \quad (3)$$

The first equality is proved in Theorem A.1 in Appendix A. Intuitively, this objective indicates we should focus on instances with higher probabilities to be labeled by human decision makers. If $e(x)$ is known exactly, we can directly select the instances by the acquisition function $e(x)\hat{\mathbb{I}}(y,\theta|x,\mathcal{D})$. It's worth noting that when $e(x) \equiv c, 0 < c \le 1$, which means human decision makers will choose to label all instances with the same probability $c$, the Naive-BALD and $e$-BALD objective is equivalent and the problem reduces to the traditional active learning setting.

Since in our setting, $e(x)$ is not known a priori, we can fit another predictive model $\hat{e}(x) = P(a=1|x,\phi)$ on the human decision dataset $\mathcal{L}$ to approximate this objective. However, in practice, $\hat{e}(x)$ may be inaccurate given a small dataset. Therefore, we need to acquire samples that are 1) likely to be labeled by the human decision makers and 2) are both informative to the machine learning model and the human discretion model update.

We consider the information gain about the human discretion response and the model parameters. Assume we use a Bayesian neural network $e_\phi(x)$ to approximate the human discretion model $e(x)$, we can define the information gain about the human discretion response and the model parameters as

$$\hat{\mathbb{I}}(a,\phi|x,\mathcal{L}) \approx -\sum_{c\in\{0,1\}}\left(\frac{1}{T}\sum_t \hat{g}_c^t\right)\log\left(\frac{1}{T}\sum_t \hat{g}_c^t\right) + \frac{1}{T}\sum_{c,t}\hat{g}_c^t\log(\hat{g}_c^t). \quad (4)$$

**Algorithm 1** Bayesian Active Learning for Selective Labeling with Instance Rejection (SEL-BALD)

---
**Input:** Initial labeled dataset $\mathcal{D}, \mathcal{L}$, pool dataset $\mathcal{D}_{\text{pool}}$, $\beta$, MC-samples $T$, budget $B$.
**Output:** Updated machine learning model $f(x)$.
**while** Budget is not used up **do**
    Select the instance $x_i$ that maximizes the acquisition function in Equation (5), (6), or (7).
    Human will decide whether to label the instance $x_i$.
    **if** the instance $x_i$ is labeled **then**
        Add $(x_i, y_i)$ to $\mathcal{D}$ and $(x_i, a_i)$ to $\mathcal{L}$.
    **else**
        Add $(x_i, a_i)$ to $\mathcal{L}$.
    **end if**
    Update the human discretion model $e_{\phi_{k+1}}(x)$ on $\mathcal{L}$.
    Update the machine learning model $f_{\theta_{k+1}}(x)$ on $\mathcal{D}$.
**end while**

---

Here $\hat{g}_c^t$ is the probability of input $x$ being labeled as $c$ under the $t$-th MC sample from the approximated posterior $q(\phi)$. We denote this objective as $\hat{\mathbb{I}}(l, \phi|x, \mathcal{L}) \approx \mathbb{I}(l, \phi|x, \mathcal{L})$. Selecting samples that maximize this acquisition function will help us to learn more effectively about the human discretion model. This joint information gain can be estimated as

**Definition 4.3** (Joint-BALD)**.**

$$\hat{\mathbb{I}}(\{a, y\}, \{\theta, \phi\}|x, \mathcal{L}, \mathcal{D}) = e(x)\hat{\mathbb{I}}(y, \theta|x, \mathcal{D}, a = 1) + \hat{\mathbb{I}}(a, \phi|x, \mathcal{L}) \tag{5}$$

We prove Equation (5) in Theorem A.2. The first term in Equation (5) is equivalent to Equation (3) and the second term is the information gain between the human discretion model and human responses. If we have the perfect knowledge about the human discretion model, then the second term equals 0 and Joint-BALD reduces to $e$-BALD. On the other hand, if we have a high epistemic uncertainty about human responses, the second term will encourage exploration to learn the human discretion model better. Once the second term becomes small (we have a good understanding of the human discretion model), Joint-BALD will automatically reduce to $e$-BALD and start exploiting.

Motivated by the upper confidence bound (UCB) principle in online regret minimization [Auer, 2002, Li et al., 2010], we can define the acquisition function as

**Definition 4.4** (Joint-BALD-UCB)**.**

$$\hat{\mathbb{I}}(\{l, y\}, \{\theta, \phi\}|x, \mathcal{L}, \mathcal{D}) = Q(\hat{e}_\phi(x), \beta)\hat{\mathbb{I}}(y, \theta|x, \mathcal{D}) + \hat{\mathbb{I}}(l, \phi|x, \mathcal{L}) \tag{6}$$

Here $\hat{Q}$ is the empirical quantile function and $\hat{Q}(\hat{e}_\phi(x_i), \beta)$ is the upper confidence bound of the probabilities (drawn using the MC samples) of $x_i$ being labeled by the human decision makers of our current best estimation. The hyperparameter $\beta$ balances the exploration-exploitation trade-off. With a larger $\beta \to 1$, the acquisition function will have more exploration of samples with higher uncertainties. Similarly, we can use Thompson Sampling [Thompson, 1933] to control the degree of exploration, which leads to the following objective:

**Definition 4.5** (Joint-BALD-TS)**.**

$$\hat{\mathbb{I}}(\{l, y\}, \{\theta, \phi\}|x, \mathcal{L}, \mathcal{D}) = q(\hat{e}_\phi(x))\hat{\mathbb{I}}(y, \theta|x, \mathcal{D}) + \hat{\mathbb{I}}(l, \phi|x, \mathcal{L}) \tag{7}$$

$q(\hat{e}_\phi(x))$ represents the empirical sample drawn from the MC samples. Each time we will random sample a set of model parameters from the posterior and select the instance with the largest information gain calculated using Equation (7). Compared to UCB, Joint-BALD-TS does not have to specify the hyperparameter. The complete algorithm for Bayesian active learning with selective labeling (SEL-BALD) is shown in Algorithm 1.

## 5 Experiments

First, we want to emphasize that in ALIR, the benefit of each method is more nuanced than traditional active learning and highly depends on human decision behavior and the underlying data distribution.

As our analysis above suggests, Naive-BALD and $e$-BALD correspond to the information gain when $e(x) \equiv c, 0 < c \leq 1$. When $c = 1$, this corresponds to the traditional active learning problem where humans always label the instance, and the efficacy of Naive-BALD and $e$-BALD have been demonstrated in existing papers [Gal et al., 2017]. We also conduct the experiment under homogeneous human decision behavior in Appendix C. In this section, we aim to show the benefit of the proposed method when the human discretion behavior is unknown and heterogeneous.

We conduct experiments on both synthetic and real-world datasets to demonstrate the effectiveness of our proposed algorithm when human discretion behavior is unknown and heterogeneous. We assume there is a cost of 1 to examine every data instance and a cost of 5 to label the instance. For the predictive model and human discretion model class, we use a Bayesian neural network and use MCdropout [Gal et al., 2017] to approximate the posterior (we set the number of MC samples as 40 in all experiments). The model architecture is a 3-layer fully connected neural network with LeakyReLU activation function. We use the Adam optimizer with a learning rate of 0.01. We run the experiments on a server with 3 Nvidia A100 graphics cards and AMD EPYC 7763 64-Core Processor. We compare our proposed methods against the Naive-BALD algorithm and the random acquisition function (RANDOM). We set $\beta = 0.75$ for Joint-BALD-UCB in all experiments. For results, we report the accuracy and number of samples acquired of the machine learning model on a subset of the test set where the instance has a positive probability to be included in the dataset. We do not consider the performance of instances that have zero probability of showing up in the data since human decision makers will always reject such decision instances and a machine learning model can have arbitrary performance on such subpopulation. The code is available at `https://github.com/ruijiang81/SEL-BALD`.

## 5.1 Synthetic Data

We use a comprehensive synthetic example to illustrate the benefit of the proposed algorithms. We generate a synthetic two-cricle dataset with 2 features. The response variable is binary and can be considered as fraud of an insurance claim. We generate 20000 data samples and downsample 90% of the data points with $x_1 \geq 0$ where $x_1$ is the second dimension of the input feature. This is to make the toy example a little more challenging for the RANDOM baseline and simulate a real-world scenario where there are a lot of insurance claims in the candidate pool that won't be investigated where active learning is necessary to reduce the potential waste of human workers' time. The resulting dataset has around 3700 samples as the training set and around 1600 samples as the test set. The visualization of the synthetic data is shown in Figure 11(a).

The qualitative results for each label acquisition method are shown in Figure 2 and the quantitative metrics are reported in Table 1. Since there are many high-risk samples that human decision makers do not label, most of the samples RANDOM selects are not labeled by the human decision maker. Similarly, Naive-BALD is highly influenced by the initial labels and human discretion behavior. If the most informative samples to the current machine learning classifier have a low probability to be labeled by humans, Naive-BALD will keep sending such instances to humans to examine and waste all the budget. The initial labels sampled are shown in Figure 11(b). Since there are not many samples when $x_1 < 0$, the human discretion model cannot estimate the correct human decisions accurately. As a result, $e$-BALD relies on the noisy estimation of the human discretion model and greedily explores the lower right region of the feature space while it is important to acquire samples from the left region in order to learn the correct decision boundary. Compared to $e$-BALD, Joint-BALD has a better exploration and labeled a few instances on the left, however, it is also influenced by the noisy estimation of $e(x)$ and focus more on the points on the right side of the sample space. Joint-BALD-UCB and Joint-BALD-TS both encourage more exploration for $e(x)$. As a result, both methods can learn the correct decision boundary in a sample-efficient way.

Finally, we show the estimated human discretion model in Figure 3. Since $e$-BALD acquires samples based on its current noisy estimation of the human discretion model, it fails to learn the correct decision boundary and maintains a wrong probability estimation about the left lower region of the feature space. While Joint-BALD has a slightly better estimation, it still focuses more on the high-probability region and has a large estimation error on the left lower region. Joint-BALD-UCB and Joint-BALD-UCB correctly recover the decision boundary with the help of sufficient exploration.

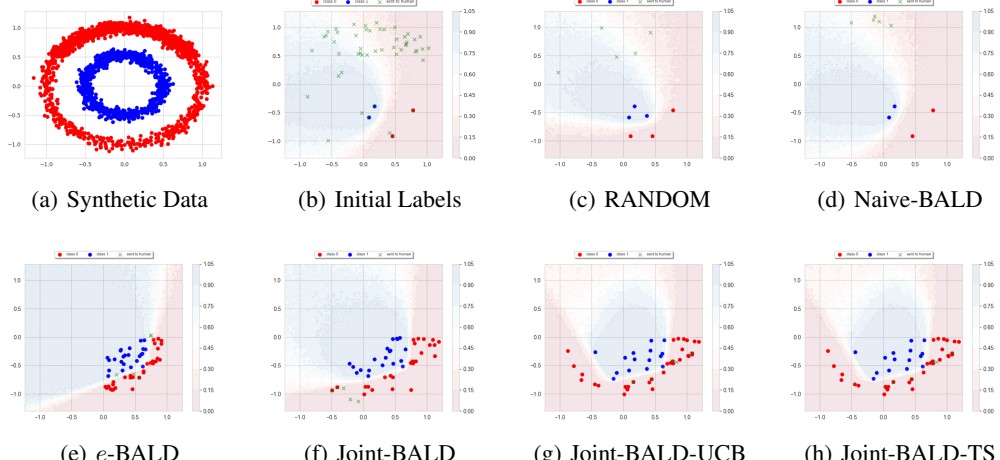

|   | (a) Synthetic Data | (b) Initial Labels | (c) RANDOM | (d) Naive-BALD |
|---|---|---|---|---|
|   | (e) $e$-BALD | (f) Joint-BALD | (g) Joint-BALD-UCB | (h) Joint-BALD-TS |

Figure 2: Qualitative Results for Synthetic Data. Figure 11(a) shows the underlying synthetic data distribution. Figure 11(b) shows the initial labels randomly acquired. Figure 2 (c)-(h) show the labeled data the corresponding learned decision boundary after spending a budget of 450. Human decision makers *always reject* to label when $x_1 > 0$. RANDOM and Naive-BALD spend most of the budget on examination since many candidate applications are in the high-risk region and have low probabilities to be labeled. In contrast, $e$-BALD selects the samples that are most likely to be labeled according to a noisy estimation of the human discretion model, which overlooks the lower left region which is important for learning the correct decision boundary.

|   | RANDOM | Naive-BALD | $e$-BALD | Joint-BALD | Joint-BALD-UCB | Joint-BALD-TS |
|---|---|---|---|---|---|---|
| Accuracy | 0.78 | 0.55 | 0.76 | 0.77 | 0.99 | 0.98 |
| Examination (%) | 0.91 | 0.95 | 0.39 | 0.52 | 0.53 | 0.50 |
| Labeling (%) | 0.09 | 0.05 | 0.61 | 0.48 | 0.47 | 0.50 |

Table 1: Quantitative Statistics for Synthetic Data. We report the accuracy and the fraction of the budget spent for examination and labeling. RANDOM and Naive-BALD spend most of the budget on examination and few labels is acquired. $e$-BALD acquires the most labels but many of them are not informative due to the noisy estimation of $e(x)$. Joint-BALD-UCB and Joint-BALD-TS balance the examination and labeling budget automatically and achieve the best accuracy.

## 5.2 Case Study: Fraud Detection

For our case study, we use the Give-Me-some-Credit (GMC) dataset [Credit Fusion, 2011]. GMC dataset is a real-world loan dataset with 10 features including age, monthly income, debt ratio, number of dependents, and a binary response variable that indicates whether the loan application will default. We balance the dataset by random undersampling and randomly choose 400 samples as the test set. We use this dataset to build a hypothetical insurance claim case. Assume each instance is an insurance claim where a human agent can investigate whether it is fraudulent. The insurance company wants to build a machine learning model to predict fraud cases in the future. Since such investigation is costly, the company wants to use an active learning method and send some selected instances to agents for potential investigation and agents have the freedom to decline the investigation request since the company does not want to waste human resources and potential profit from acquiring these samples.

**Human Behavior Model:** For the human discretion model $e(x)$, we assume human decision makers will have the following acceptance probabilities:

$$e(x) = \begin{cases} 0.3 & \text{High Debt and Low Income} \\ 0.9 & \text{High Debt and High Income} \\ 0 & \text{Low Debt.} \end{cases}$$

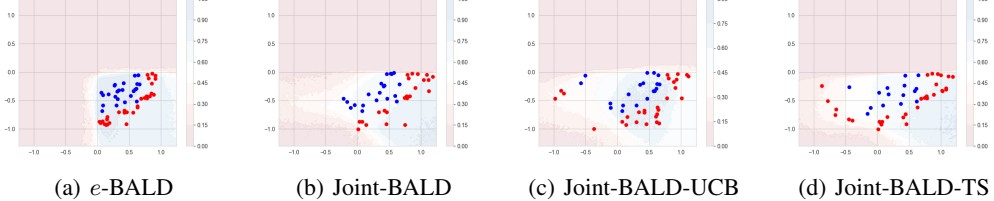

| (a) $e$-BALD | (b) Joint-BALD | (c) Joint-BALD-UCB | (d) Joint-BALD-TS |

Figure 3: Estimated Human Discretion Behavior for Synthetic Data. We show the decision boundary for the estimated human discretion behavior for each method after spending a budget of 450. $e$-BALD underestimates the lower left region of the feature space due to the lack of exploration. Joint-BALD-UCB and Joint-BALD-UCB recover the correct decision boundary.

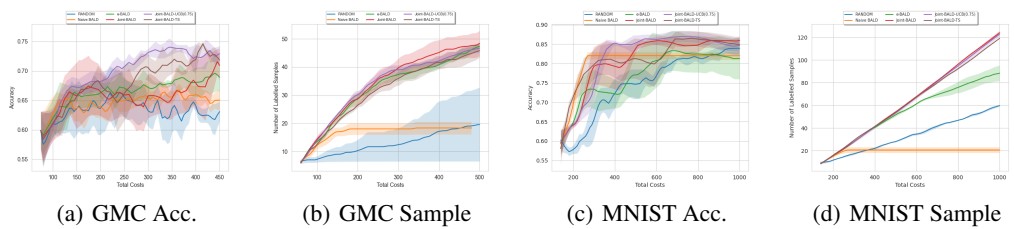

| (a) GMC Acc. | (b) GMC Sample | (c) MNIST Acc. | (d) MNIST Sample |

Figure 4: Results for GMC and MNIST. We report the accuracy and number of samples labeled for both datasets. RANDOM and Naive-BALD spend most of the budget on examination since they do not consider human discretion behavior. On both datasets, Joint-BALD-UCB and Joint-BALD-TS demonstrate robust performance across different budgets.

A detailed discussion about the initial human discretion model can be found in Appendix D. This simulates the selective labels setup where the biased human decision makers are skeptical about insured customers with high debt and these customers are more likely to be investigated.

We report the accuracy and number of samples acquired for each method in Figure 4(a) and Figure 4(b) respectively. The results are averaged over 3 runs with a query size of 10, 50 randomly examined instances initially, and a budget of 450. Compared to other methods, RANDOM and Naive-BALD acquire the least samples given the same budget since both methods do not consider the human discretion behaviors, which leads to suboptimal model performance. $e$-bald has a small benefit over all other methods when the total budget is small, which is expected when the initial estimated human discretion model is useful since $e$-BALD greedily acquires samples based on the current best estimation. However, this effect soon disappears and $e$-bald has a worse performance compared to Joint-BALD-UCB and Joint-BALD-TS which have a better exploration of $e(x)$ for a larger budget.

### 5.3   MNIST

We also examine each active learning method on a high-dimensional dataset MNIST [LeCun, 1998]. MNIST is a large handwritten digit dataset with 28x28 pixels and 10 classes. The test set size is 10000 and we use 10000 samples in the training set as $\mathcal{D}_{\text{pool}}$. We report the accuracy and number of samples acquired for each active learning method on the test set. The results are averaged over 3 runs with a query size of 20, 100 randomly examined instances initially, and a budget of 1000. For the human discretion behavior, we assume $P(a = 1|x) = 0$ for all classes 0-6, and human decision makers have the probabilities of 0.3, 0.3, 0.9 to label the instances with classes 7, 8, 9 respectively. This also corresponds to the realistic setting where many insurance claims will not be investigated and human discretion behavior is different for different instances.

The results are shown in Figure 4(c) and Figure 4(d) respectively. Similarly to our previous findings, baselines like RANDOM and Naive-BALD struggle to acquire samples humans will label, and waste most of the budget on examination. For high-dimensional data, the performance of $e$-BALD appears worse in the beginning potentially because the propensity score is harder to estimate for high-dimensional data [D'Amour et al., 2021]. On the other hand, Joint-BALD-UCB and Joint-BALD-TS

are able to learn the human discretion model better through exploration, and lead to a larger number of samples labeled and better model performance.

## 5.4 Additional Experiments

**Changing Human Behaviors:** While we motivate the Active Learning with Instance Rejection problem in the cold-start setting, the ALIR problem can still happen when human behavior changes, thus we need to acquire new labels to better model the underlying function and the human discretion model. We offer an illustrative example of ALIR under changing human behaviors in Appendix B. In the example, we assume the company starts to use a machine learning model to advise human decision makers, thus leading to a change in the human discretion behavior. We further demonstrate that the ALIR problem still exist when human behaviors change and the proposed methods in the paper can also be applied in such settings.

**More Real-World Datasets:** We compare the proposed methods with baselines on Fashion MNIST [Xiao et al., 2017], CIFAR-10 [Krizhevsky, 2009], Adult [Becker and Kohavi, 1996] and Mushroom [mus, 1981] datasets in Appendix E. We find similar findings that the proposed methods improve over Naive-BALD and RANDOM under the selective labels setting.

**Other Human Behavior Model:** In addition to the piecewise linear function used to simulate the human behaviors, we compare with a Logistic Regression based human behavior model in Appendix F. We randomly sample a few positive and negative samples from the datasets to train a Logistic Regression model and use it to simulate human responses. The results are similar to our results presented in the main paper.

**Other Uncertainty Metric:** While we focus on improving the BALD method in this paper, it is possible to extend the proposed method for these metrics by replacing the information gain term with another uncertainty term such as the entropy. We demonstrate that in Appendix G. We implement Joint-Entropy-UCB that replaces the information gain with the entropy measure on the synthetic data. Joint-Entropy-UCB can also improve the traditional entropy objective in this case.

## 6 Conclusion and Future Work

In this paper, we study the Active Learning with Instance Rejection (ALIR) problem, which is an active learning problem that considers the human discretion behavior for high-stakes decision making problems. We propose new active learning algorithms $e$-BALD, Joint-BALD, Joint-BALD-UCB, and Joint-BALD-TS using deep Bayesian active learning for selective labeling (SEL-BALD) to address the ALIR problem and validate the effectiveness of our proposed algorithm on both synthetic and real-world datasets. With unknown and heterogeneous human behavior, Joint-BALD-UCB and Joint-BALD-TS often lead to robust model performance improvement across different budgets.

However, there are several limitations to our work. First, we consider all humans as a single entity and assume they have the same discretion model which is an average of all humans. In practice, different human decision makers may have different discretion models. Second, it is also promising to use semi-supervised learning [Grandvalet and Bengio, 2004] to train the predictive model while it brings additional challenges to the ALIR problem. In addition, $e(x)$ may not be accurately modeled which could affect the performance of the proposed algorithm. One potential issue is the misspecification of the model class used to represent human behavior. For instance, if a linear model like Logistic Regression is applied to a fundamentally non-linear human behavior pattern, this could introduce estimation bias into the proposed AL method. To mitigate this, non-parametric models could be employed, as they are less prone to such misspecification bias. Last, it is also worth noting that in certain applications like recidivism prediction or organ transplantation, prioritizing instances to be labeled may have significant fairness implications, therefore our method may not apply. However, the ALIR problem still exist in many real-world problems such as tax audit or fraud detection. It would be interesting to study the fairness concerns in ALIR. We believe these bring promising future research opportunities and leave these as future work.

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

# A  Proofs

**Theorem A.1** (e-BALD). *Assume when $a = 0$, the observation is a fixed non-informative value $y_0$ (e.g., NA). If the human decision maker has a probability $e(x)$ to label the instance, then*

$$\mathbb{I}(y; \theta | x, \mathcal{D}) = e(x)\mathbb{I}(y; \theta | x, \mathcal{D}, a = 1).$$

*Proof.* Let $e(x) = p(a = 1 | x)$.

$$\mathbb{I}(y; \theta | x, \mathcal{D})$$
$$= \mathbb{E}_{y,\theta|x,\mathcal{D}} \log \frac{p(y|\theta, x, \mathcal{D})}{p(y|x, \mathcal{D})}$$
$$= e(x)\mathbb{E}_{y,\theta|a=1,x,\mathcal{D}} \log \frac{p(y|\theta, x, \mathcal{D}, a = 1)}{p(y|x, \mathcal{D}, a = 1)} + (1 - e(x))\mathbb{E}_{y,\theta|a=0,x,\mathcal{D}} \log \frac{\delta_{y_0}(y)}{\delta_{y_0}(y)}$$
$$= e(x)\mathbb{I}(y; \theta | x, \mathcal{D}, a = 1).$$

The second equality is because of the law of total expectation. $\square$

**Theorem A.2** (Joint-BALD). *Under the condition of Theorem A.1,*

$$\mathbb{I}(\{a, y\}; \{\theta, \phi\} | x, \mathcal{L}, \mathcal{D}) = e(x)\mathbb{I}(y; \theta | x, \mathcal{D}, a = 1) + \mathbb{I}(a; \phi | x, \mathcal{L}).$$

*Proof.* Recall that $\mathcal{L} = \{x_i, a_i\}_{i=1:n}$ and $\mathcal{D} = \{x_i, y_i\}_{a_i=1}$.

$$\mathbb{I}(a, y; \theta, \phi | x, \mathcal{L}, \mathcal{D})$$
$$= \mathbb{I}(a; \theta, \phi | x, \mathcal{L}, \mathcal{D}) + \mathbb{I}(y; \theta, \phi | a, x, \mathcal{L}, \mathcal{D})$$
$$= \mathbb{I}(a; \theta, \phi | x, \mathcal{L}) + \mathbb{I}(y; \theta, \phi | a, x, \mathcal{D})$$
$$= \mathbb{E}_{a,\phi|x,\mathcal{L}} \log \frac{p(a|\phi, x, \mathcal{L})}{p(a|x, \mathcal{L})} + \mathbb{E}_{a|x}\mathbb{E}_{y,\theta|a,x,\mathcal{D}} \log \frac{p(y|\theta, x, a, \mathcal{D})}{p(y|x, a, \mathcal{D})}$$
$$= \mathbb{I}(a; \phi | x, \mathcal{L}) + e(x)\mathbb{E}_{y,\theta|a=1,x,\mathcal{D}} \log \frac{p(y|\theta, x, a = 1, \mathcal{D})}{p(y|x, a = 1, \mathcal{D})}$$
$$\qquad + (1 - e(x))\mathbb{E}_{y,\theta|a=0,x,\mathcal{D}} \log \frac{\delta_{y_0}(y)}{\delta_{y_0}(y)}$$
$$= \mathbb{I}(a; \phi | x, \mathcal{L}) + e(x)\mathbb{E}_{y,\theta|a=1,x,\mathcal{D}} \log \frac{p(y|\theta, x, a = 1, \mathcal{D})}{p(y|x, a = 1, \mathcal{D})}$$
$$= \mathbb{I}(a; \phi | x, \mathcal{L}) + e(x)\mathbb{I}(y; \theta | x, \mathcal{D}, a = 1).$$

The first equality is by the chain rule of mutual information; the second equality is because the observation of all $a$ is in $\mathcal{L}$ and all the observation of $y$ is in $\mathcal{D}$; the third equality is because the distribution of $y$ and $a$ are independent of $\phi$ and $\theta$, respectively. $\square$

# B    Extensions on AI Advising/Changing Human Behavior

While we motivate the Active Learning with Instance Rejection problem in the cold-start setting in the main paper, the Active Learning with Instance Rejection can still happen when human behavior changes, thus we need to acquire new labels to better model the underlying function and the human discretion model. Here we offer an illustrative toy example of ALIR under changing human behaviors. In this toy example, we assume the company starts to use a machine learning model to advise human decision makers, thus leading to human discretion behavior change.

We use a synthetic example to illustrate the AI advising setting We generate a synthetic two-moon dataset with 2 features. The response variable is binary and can be considered as whether an insurance is fraudulent. We generate 20000 data samples and downsample 90% of the data points with $x_0 \leq 6$ where $x_0$ is the first dimension of the input feature. This simulates a real-world scenario where there are a lot of valid insurance claims in the candidate pool where active learning is necessary to reduce the potential waste of human workers' time. The resulting dataset has around 3700 samples as the training set and around 1600 samples as the test set. The visualization of the synthetic data is shown in Figure 5(a).

**Initial Human Behavior Model:** For the initial human discretion model $e_0$, we assume human decision makers will only investigate claims with $3 \leq x_0 \leq 6$ and follow the acceptance probabilities below:

$$h_0(x) = \begin{cases} 0.6 & 3 \leq x_0 \leq 6 \text{ and } x_1 > 3 \\ 0.4 & 3 \leq x_0 \leq 6 \text{ and } x_1 \leq 3 \\ 0 & \text{otherwise.} \end{cases} \tag{8}$$

This simulates the selective labels setup where initially human decision makers will only investigate claims of a certain subpopulation, and the machine learning model fitted on this subpopulation may not generalize well to the whole population even with ample data. Here the human decision makers have a higher probability to investigate fraud cases.

**Updated Human Behavior Model:** We use the initial human behavior to collect 200 samples with instance rejection, then we train a Bayesian neural network $f_0(x)$ on the initial labeled dataset $\mathcal{D}$ and deploy it to assist human decision makers. After the machine learning model is deployed, we assume the human decision makers will have a new behavior model $e(x)$ that follows the acceptance probabilities below:

$$h(x) = \begin{cases} 0.6\rho + (1-\rho)f_0(x) & 3 \leq x_0 \leq 6 \text{ and } x_1 > 3 \\ 0.2\rho + (1-\rho)f_0(x) & 3 \leq x_0 \leq 6 \text{ and } x_1 \leq 3 \\ 0\rho + (1-\rho)f_0(x) & x_0 \leq 3 \\ 0 & \text{otherwise.} \end{cases}$$

Here $\rho$ is an algorithmic-aversion hyperparameter that controls the level of trust in the machine learning model. When $\rho = 0$, the human decision makers will fully trust the machine learning model. When $\rho = 1$, the human decision makers will ignore the machine learning model and follow the initial human behavior model. We set $\rho = 0.5$ to simulate a partial adoption of the machine learning model. However, for the high-risk region $x_0 > 6$, the human decision makers will never investigate no matter what the machine learning model advises.

Since the updated human behavior model may have higher label probabilities for the cases at certain regions (e.g., $x_0 \leq 3$), we can use different active learning strategies to acquire new samples to improve the machine learning model's generalizability. We test Naive-BALD, RANDOM, and Joint-BALD-UCB on this synthetic dataset for illustration. We set the query size as 20 and the number of acquisition steps as 10. For batch acquisition with query size $K$, we use the greedy strategy to select the $K$ samples with the largest acquisition function value.

First, we visualize the labeled data after 3 and 9 steps of active learning in Figure 5. We can see that Joint-BALD-UCB selects the samples that are likely to be labeled by the human decision makers and are both informative to the machine learning model and the human discretion model update. In contrast, Naive-BALD selects the samples that are informative to the machine learning model but may

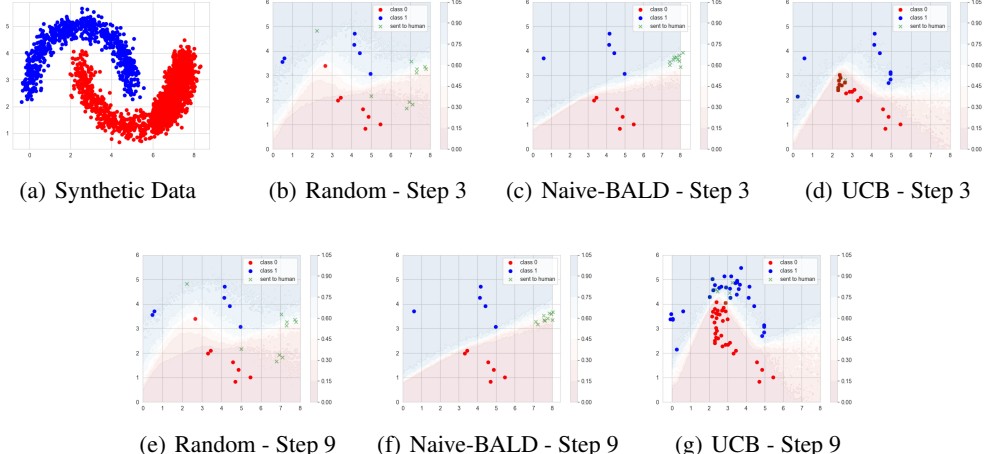

(a) Synthetic Data  (b) Random - Step 3  (c) Naive-BALD - Step 3  (d) UCB - Step 3

(e) Random - Step 9  (f) Naive-BALD - Step 9  (g) UCB - Step 9

Figure 5: Synthetic Data. Figure 5(a) shows the underlying synthetic data distribution. Figure 5 (b)-(d) and (e)-(g) shows the labeled data after 3 and 9 steps of active learning. Human decision makers *always reject* to label when $x_0 > 6$. BALIR selects the samples that are likely to be labeled by the human decision makers and are both informative to the machine learning model and the human discretion model update. In contrast, BALD selects the samples that are informative to the machine learning model but may not be labeled by the human decision makers. RANDOM selects the samples randomly and since many candidate applications are in the high-risk region, it also has a low probability to select the samples that will be labeled by the human decision makers.

not be labeled by the human decision makers. In this case, Naive-BALD got stuck in the high-risk region $x_0 > 6$ where human decision makers will always reject the loan applications, thus leading to no improvement for the machine learning model. RANDOM selects the samples randomly and since many candidate applications are in the high-risk region, it also has a low probability of selecting the samples that will be labeled by the human decision makers. Joint-BALD-UCB successfully recovers the correct decision boundary. This example aims to demonstrate that the ALIR problem still exist when human behavior change and the proposed methods in the paper can also be applied in such settings.

## C   Homogeneous Human Behavior

In this section, we show how different active learning methods perform under homogenous human behaviors. As demonstrated in the main paper, when $e(x) \equiv c, 0 < c \leq 1$, Naive-BALD corresponds to the correct information gain and we expect all the proposed methods to perform similarly.

We use the same experimental setup in Section 5.3 with the MNIST database and change the human decision behavior as $e(x) \equiv 0.8$. The model accuracy and number of samples labeled are reported in Figure 6(a) and Figure 6(b) respectively. Naive-BALD and $e$-BALD also have similar model performance compared to other SEL-BALD methods we proposed in the paper.

## D   Human Behavior Model in Case Study

The human discretion model used in the case study is

$$
e(x) = \begin{cases} 0.3 & \text{Debt} > \hat{Q}(\text{Debt}, 0.7) \text{ and Income} \leq \hat{Q}(\text{Income}, 0.7) \\ 0.9 & \text{Debt} > \hat{Q}(\text{Debt}, 0.7) \text{ and Income} > \hat{Q}(\text{Income}, 0.7) \\ 0 & \text{Debt} \leq \hat{Q}(\text{Debt}, 0.7). \end{cases}
$$

where $\hat{Q}(\cdot, \alpha)$ is the empirical $100 \times \alpha$ quantile of the metric. This simulates the selective labels setup where human decision makers are skeptical about insured customers who have high debt.

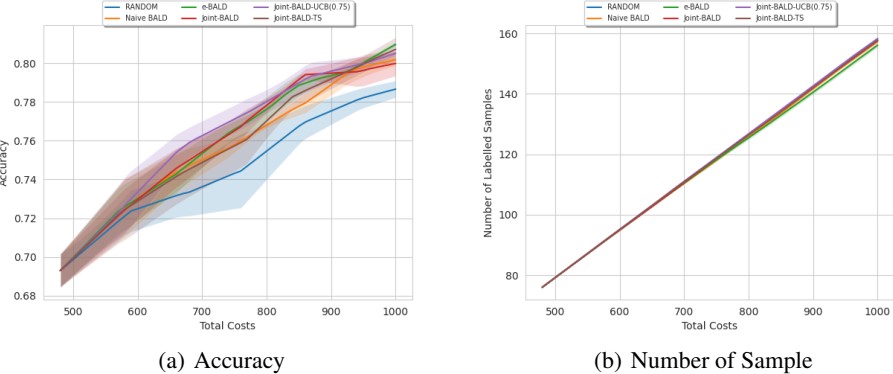

(a) Accuracy  (b) Number of Sample

Figure 6: Results for MNIST with Homogeneous Human Behavior. We report accuracy and number of samples labeled. All the proposed methods have similar model performance with homogeneous human discretion behavior across different budgets.

# E   Experiments with More Datasets

We compare the proposed methods with baselines on Fashion MNIST [Xiao et al., 2017], CIFAR-10 [Krizhevsky, 2009], Adult [Becker and Kohavi, 1996] and Mushroom [mus, 1981] datasets in this section. For Fashion MNIST and CIFAR-10, we use the same human behavior model as the one used in MNIST. For Mushroom, we assume humans have the following probability to label the instances: 0 when $x_1 < 0$; 0.3 when $x_1 > 0$ and $x_4 > 0$; 0.9 otherwise. For Adult, we assume humans have the following probability to label the instances: 0 when $x_0 > Q_{0.5}(x_0)$; 0.3 when $x_0 \leq Q_{0.5}(x_0)$ and $x_1 > Q_{0.5}(x_1)$; 0.9 otherwise.

The results for the accuracy and the number of samples labeled are shown in Figure 7 and Figure 8, respectively. We find similar qualitative conclusions that Joint-BALD variants often produce better results than Naive-BALD and RANDOM.

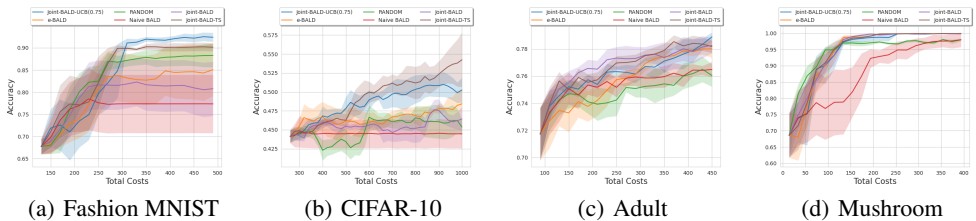

(a) Fashion MNIST  (b) CIFAR-10  (c) Adult  (d) Mushroom

Figure 7: Experimental Results (Accuracy) for Additional Datasets. We report the accuracy for each cost on different additional datasets. Joint-BALD variants often produce better results than Naive-BALD and RANDOM across different costs. Fashion-MNIST and CIFAR-10 use the same setting as MNIST in the main paper.

# F   Experiments with ML-based Human Behavior Model

In addition to the piecewise linear human behavior model we used, we compare with a Logistic Regression based human behavior model. We randomly sample 3 positive and negative samples from the datasets to train a Logistic Regression model and use it to simulate human responses. Humans have a higher probability to label the instances if the predicted probabilities from the Logistic Regression are larger and humans will reject to label when the predicted probabilities are low.

The results with Adult dataset is shown in Figure 9(a) and Figure 9(b) and results with Mushroom dataset is shown in Figure 10(a) and Figure 10(b). The proposed methods can significantly outperform baselines like Naive-BALD. We also note RANDOM sometimes is a strong baseline (e.g., in the

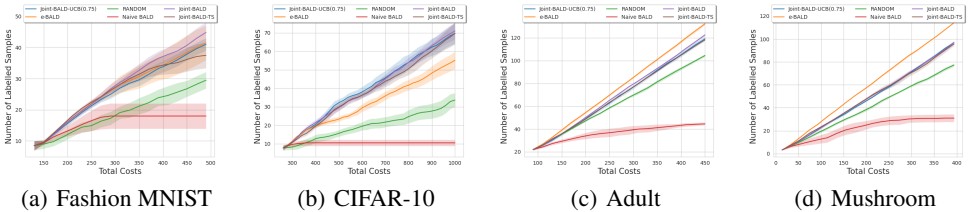

| (a) Fashion MNIST | (b) CIFAR-10 | (c) Adult | (d) Mushroom |

Figure 8: Experimental Results (Number of Samples) for Additional Datasets. We report the number of samples labeled for each cost on different additional datasets. Naive-BALD and RANDOM label much less samples than the Joint-BALD variants and $e$-BALD often labels the most labels.

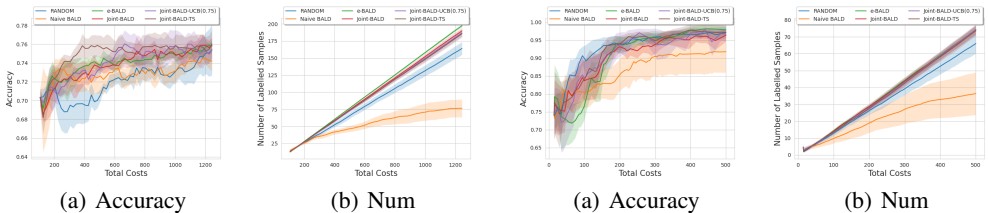

| (a) Accuracy | (b) Num | (a) Accuracy | (b) Num |

Figure 9: ML Human Behavior Model (Adult)     Figure 10: ML Human Behavior (Mushroom)

Mushroom dataset) in the batch active learning setting as documented in the previous work [Kirsch et al., 2019].

# G   Experiments with Other Uncertainty Metric

While we mainly focus on improving the BALD objective in this paper, our methods can potentially be generalized by replacing the mutual information with other metrics in the objective. For example, we can replace the information gain with other uncertainty measures such as entropy. A similar Joint-Entropy-UCB objective can be written as

$$\text{Ent}(\{l, y\}, \{\theta, \phi\}|x, \mathcal{L}, \mathcal{D}) = Q(\hat{e}_\phi(x), \beta)\text{Ent}(y, \theta|x, \mathcal{D}) + \text{Ent}(l, \phi|x, \mathcal{L}) \quad (9)$$

As shown in Figure 11, we implement Joint-Entropy-UCB with entropy measure on the synthetic data. Joint-Entropy-UCB can also improve the traditional entropy objective in this case.

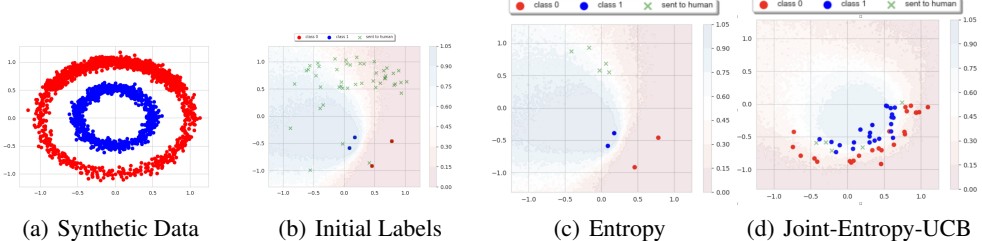

| (a) Synthetic Data | (b) Initial Labels | (c) Entropy | (d) Joint-Entropy-UCB |

Figure 11: Extension on other uncertainty-based metric: Qualitative Results for Synthetic Data after spending a budget of 450. Our method can also help improve the performance of other uncertainty-based active learning metric like Entropy.

