# OpenReview forum: "SEL-BALD: Deep Bayesian Active Learning with Selective Labels"
_NeurIPS.cc/2024/Conference — NeurIPS 2024 poster_

### Official Review · Reviewer_dpcV · 2024-06-13

**Soundness:** 3
**Presentation:** 4
**Contribution:** 4
**Rating:** 8
**Confidence:** 4

**Summary:**

This paper explores the setting where a learner has a label budget that can be used to acquire labeled data from a user. However, differently from traditional active learning approaches, the paper assumes that the user may not want/be able to label some examples. That is, for specific (limited) examples, the user returns a label when queried, while for other examples it does not return anything. The key challenge of the paper is how to motify traditional query strategies to account for this new possibility of not getting any label from the user. The paper solves this challenge by modeling the user through a Bayesian network and including two additional terms in the AL score function: the first term scales the potential gain obtained by labeling an instance by the probability that the instance would get labeled when queried, while the second represents the gain obtained for modeling the user (i.e., where the user labels the data).

**Strengths:**

### Main Strengths

1. The problem setting is realistic, unexplored, and yet relevant for industry use cases: after all, some experts may not want, may not be allowed, or even may not be able to provide labels.

2. The paper is clearly written: claims are well supported, the notation is well detailed, and the flow is easy as well as the structure.

3. The proposed approach is reasonable and meaningful: the paper proposes multiple methods (such that one is the generalization of the other) and shows empirically how they perform. The Bayesian Active Learning with Disagreement approach is still quite used - despite being more than 10 years old. Thus, extending BALD to the new setting can have a positive impact on several industrial use cases.

4. The paper is technically sound and highlights whenever a quantity is an estimate and when is the true value.

**Weaknesses:**

### Main Weaknesses:

1. The experiments include MNIST, which is a little bit outdated and nowadays considered a simple dataset. I'd recommend using more elaborated multiclass datasets (e.g., SVHN, Waterbirds, ...). Using more common datasets would strengthen the work.

2. Some literature study might be missing: learning to defer is an area where a model defers the prediction to a human whenever the human is more likely to provide a correct prediction, whereas the model is preferred in other cases. Thus, this research direction also investigates how to model a human. Also, learning to reject is a similar area, and [1] uses active learning assuming that the user may not provide some labels. Finally, there is a connection with PU Learning (Positive and Unlabeled Learning), where the propensity score named e(x) represents the probability of labeling an instance. Including a brief connection to these areas would improve the quality of the paper.

[1]: https://www.ijcai.org/proceedings/2020/0403.pdf

**Questions:**

No questions.

**Limitations:**

Limitations are clearly discussed in Section 6.

---

> ### Author Rebuttal · Authors · 2024-08-06
>
> We're encouraged by the reviewer's positive comments on the novelty of the problem and our method contribution. Below, we address the concerns raised in your review.
>
> - **Datasets:** Thanks for the suggestion. We followed your recommendation and added additional experimental results on Fashion MNIST, CIFAR-10, and two UCI datasets (Adult and Mushroom). The results are included in Figures 2 and 3 in the one-page PDF response. We find similar qualitative conclusions that Joint-BALD variants often produce better results than Naive-BALD and RANDOM across different costs.  Please refer to **Response to All** and the attached PDF for details.
>
>  - **References:** Thanks for the great references! Indeed, learning to reject and learning to defer (adaptive learning to reject) are closely related to our work. These works consider how to select AI or the human at testing time to decide on an instance while our paper focuses on how to design the active learning strategy with human discretion behaviors. [Perini et al. 2020] studies how to estimate the class prior with a given active learning strategy in the PU learning setup. PU learning also considers a partially observed label setting similar to ours. We will discuss the relationships with these literature in detail in the revised manuscript.
>
> We appreciate your insightful suggestions and have addressed it thoroughly in our revised manuscript to enhance the overall quality of the work. Please do let us know if you have any further concerns, or if this adequately addresses all the issues that you raised with the paper.

---

> > ### Comment · Reviewer_dpcV · 2024-08-11
> >
> > Dear authors,
> >
> > Thanks for addressing my points. As stated in my review, I believe the paper is in a good shape and provides good contribution. Although the other reviewers highlighted some weakness, the pros of accepting the paper outnumber the cons, in my view. Thus, I recommend acceptance.

---

> > > ### Author Response · Authors · 2024-08-12
> > > **Thank You**
> > >
> > > Thank you so much for being positive about our work. We will include your valuable feedback in the revised version.

---

### Official Review · Reviewer_KKqr · 2024-07-10

**Soundness:** 3
**Presentation:** 3
**Contribution:** 3
**Rating:** 5
**Confidence:** 4

**Summary:**

This paper presents a direct extension of BALD sampling criterion to address the need for active annotation rejection. The method assumes a rejection distribution over the candidate data samples in pool-based active learning and proposes an active sampling strategy that jointly considers the BALD informativeness and rejection cost.

**Strengths:**

1.The problem studied in this paper can be considered an instance of cost-aware active learning. The challenge of having samples that are difficult or even impossible to label is a very practical issue in active learning. This paper provides a valuable contribution to the field of active learning by explicitly estimating the rejection function, which offers better guidance for active learning and is beneficial for improving data efficiency.

2.The rejection component proposed in this paper is naturally integrated into Bayesian sampling. The final hybrid sampling criterion revolves closely around information gain and uncertainty reduction, which I find to be reasonable and effective. Approaching active sampling from a Bayesian perspective also makes it more interpretable.

3.The paper is written in a concise and fluent manner, with high readability. The core method is introduced clearly and is easy to understand.

**Weaknesses:**

1.The core of this paper's design is the human discretion function e(x), but there are some unclear aspects regarding this function in my view. First, the approximation of e(x) seems to be achieved through an additional Bayesian network, but the detailed features of this Bayesian network are omitted in the paper. As the authors mention, given the small sample nature of active learning, the estimation of e(x) is likely to have an underfitting problem. However, the two solutions proposed by the authors in line 171 do not seem to address this issue adequately, and the authors should provide more explanation on this. Additionally, since e(x) is estimated by a Bayesian network, it should be involved in the computation in the form of some posterior distribution. However, in the examples given in the experimental results section, the true e(x) is presented as an unnormalized piecewise function. Therefore, using a posterior to approximate an e(x) that is not a valid density function raises questions about its effectiveness and limitations. I believe this is directly related to the deeper reason for label rejection, i.e., the analytical form of e(x). If the authors cannot theoretically prove that the proposed method generally works for all types of rejection reasons, they should at least compare the model's prediction of rejection (instead of the final overall sampling score) with several typical true rejection distributions in the experiments.
2.As a cost-aware active learning method, I find the experimental section of this paper somewhat weak. The real-world dataset used does not record the true rejection behavior of human annotators, which means, to some extent, that the experiments on MNIST are still synthetic. The true reasons for human annotators rejecting to label might be much more complicated than the prototype e(x) used in this paper. Therefore, I believe the evaluation of this specific problem requires a dataset with actual recorded human annotator rejections.

**Questions:**

-Would it be possible to include a comparison of the model's rejection predictions against typical true rejection distributions in your experiments?- Should the first term in eq 4 be negative?
-The constant c in section 4.2 seems to differ from the c defined in section 3.1. You might want to use a different notation.

---

> ### Author Rebuttal · Authors · 2024-08-06
>
> We are delighted to learn your acknowledgment of our Bayesian approach to a practical problem in active learning.  We address your points of concern in detail below.
>
> - **W1 (Estimating $e(x)$):**  (i) We now add more implementation details of the Bayesian network $e_\phi(x)$ in the appendix.  In the simulation, we use the MC dropout as a Bayesian Approximation [Gal & Ghahramani 2016] (we set the number of MC samples as 40 in all experiments). The model architecture is a 3-layer fully connected neural network with LeakyReLU activation function..  (ii) underfitting problem: in L171, we propose to query data points that are informative to the human discretion model in the second point, which aims to address the underfitting issue. More specifically, in Def 4.3-4.5, we include an additional term that quantifies the information gain on the human discretion model to help learn $e(x)$ better. In addition, we propose the UCB and TS variants to guide our label acquisition using the underfitted $e(x)$. As shown in Figure 3, the underfitted $e(x)$ will indeed harm methods like $e$-BALD but our proposed method can get a better performance. (iii) We agree with the reviewer that the true $e(x)$ is unnormalized, i.e. $\int e(x) dx \neq 1$. The discretion model $e_\phi(x) = p(a=1 | x, \phi)$ is a function mapping from the domain $\mathcal{X}$ of $x$ to any value in $[0,1]$, which also does not require normalization over $x$. The Bayesian computation yields the posterior $p(\phi | \mathcal{D})$ of the parameters $\phi$ instead of $e(x)$. So our $e_\phi(x)$ can be applied to approximate any function $\mathcal{X} \to [0,1]$.
>
> - **Datasets and Human Behavior Models:** Thanks for the suggestion. We added additional experimental results on Fashion MNIST, CIFAR-10, and two UCI datasets (Adult and Mushroom). The results are included in Figures 2 and 3 in the one-page PDF response. We find similar qualitative conclusions that Joint-BALD variants often produce better results than Naive-BALD and RANDOM across different costs. Though selective labeling is prevalent in industry use cases, we hope you can understand that it is challenging for us to find public real-world data with recorded human decision-maker judgments on such applications, so we choose to vary the human behavior model in the paper so that we can get a more detailed comparison and insights between different methods.
>
> Please find the results for additional datasets and human behaviors in **Response to All** and the attached PDF.
>
> - **True/Predicted Human Behavior:** We kindly note that Figure 3 in the main paper demonstrates the comparison of $e(x)$ and the estimated $\hat{e}_\phi(x)$ for the synthetic data. We will address the typos and notations in the revised version.
>
> We appreciate your insightful suggestions and have addressed it thoroughly in our revised manuscript to enhance the overall quality of the work. Please do let us know if you have any further concerns, or if this adequately addresses all the issues that you raised with the paper.

---

> > ### Comment · Reviewer_KKqr · 2024-08-08
> > **Thank you for the rebuttal**
> >
> > Thank you for the detailed rebuttal. The additional implementation details of the Bayesian network and the steps taken to address underfitting provide the necessary clarification regarding the estimation of e(x). The use of MC dropout and the model architecture explanation help in understanding the approach better. Including information gain and introducing UCB and TS variants further address my concerns. Regarding the experimental section, the addition of datasets like Fashion MNIST, CIFAR-10, and two UCI datasets strengthens the evaluation. While real-world datasets with recorded human rejections are challenging to obtain, your varied human behavior models offer valuable insights. Overall, while the paper's novelty and theoretical contributions are modest, the provided clarifications and additional experiments adequately address my concerns. I am inclined to maintain a positive rating as borderline accept.
> > Thanks!

---

> > > ### Author Response · Authors · 2024-08-12
> > > **Thank You**
> > >
> > > Thank you so much for being positive about our work. We will incorporate all your inputs into the final version. Thank you for your time and effort in helping with the review of our paper.

---

### Official Review · Reviewer_fApQ · 2024-07-11

**Soundness:** 3
**Presentation:** 2
**Contribution:** 3
**Rating:** 5
**Confidence:** 4

**Summary:**

The paper introduces the Active Learning with Instance Rejection (ALIR) problem, addressing the issue of selective labeling where human discretion impacts the labeling process. In particular, humans might not always provide a label for a point returned by active learning; this abstention needs to be modeled explicitly. The authors propose new algorithms under the SEL-BALD framework to improve model performance using a human discretion model. The paper demonstrates the effectiveness of these algorithms through experiments on both synthetic and real-world datasets.

**Strengths:**

- The paper's focus on Active Learning with Instance Rejection (ALIR) focusses on a real-world problem where human discretion plays a critical role in the labeling process. This assumption seems to be realistic and well-motivated.
- The introduction of the SEL-BALD (Selective Bayesian Active Learning by Disagreement) framework effectively balances the dual objectives of selecting informative instances and accommodating human labeling behavior.
- The empirical results show the benefits of considering human discretion in active learning. This underscores the practical advantages of the proposed method(s).
- The paper is well-structured and clearly written.

**Weaknesses:**

- The integration of Bayesian neural networks for modeling both machine learning tasks and human judgment significantly increases computational complexity. These Bayesian models are computationally demanding to train.

- While the paper includes experiments with both synthetic and real-world datasets, the scope of real-world applications examined is relatively limited. Critical domains such as medical diagnostics or financial fraud detection present unique challenges. A more comprehensive evaluation across diverse applications would better illustrate the versatility and reliability of the approach.

- The paper's assumption that human discretion can be accurately modeled using predictive tools like Bayesian neural networks may be overly optimistic. Real-world human decision-making is influenced by numerous complex and often unpredictable factors. The efficacy of these models can vary significantly across different contexts. Accurate modeling of human discretion typically requires extensive, high-quality historical data on human decisions, which may not always be available. In cases where data is limited, noisy, or biased, the human discretion model's performance could be significantly compromised. Furthermore, human decision-making processes can change over time due to regulatory updates, new information, or shifts in organizational policies. The study does not address how the human discretion model would adapt to such changes, potentially leading to outdated or inaccurate predictions.

- The legibility of figure legends is consistently poor due to their small size. This aspect requires significant improvement for better readability and comprehension.

- Definitions 4.1 through 4.5 would benefit from more detailed explanations accompanying the equations, providing context and clarification for the mathematical formulations presented

**Questions:**

- Can you expand the techniques used to train the human discretion model?
- How can potential biases introduced by assumptions on the human discretion model be addressed?
- Is $c_e$ part of $c_l$?

**Limitations:**

See weaknesses above.

---

> ### Author Rebuttal · Authors · 2024-08-06
>
> Thank you very much for your positive comments on our problem, method, and writing. Nevertheless, we understand there are concerns impacting the evaluation.
>
> To address these:
>
> - **Computational Complexity:** In this paper, we mainly focus on improving the BALD method, therefore the baseline models are also Bayesian models that share the same computational complexity. In addition, we use the MC dropout as a Bayesian Approximation [Gal & Ghahramani 2016] which is not computationally demanding compared to a deterministic model with the same architecture. For other traditional AL metrics that work for simpler models such as Entropy, it is also possible to extend the proposed method for these metrics by replacing the information gain term with the uncertainty term. We demonstrate that in Figure 1 in the response PDF, we implement Joint-Entropy-UCB with entropy measure on the synthetic data. Joint-Entropy-UCB can also improve the traditional entropy objective in this case. We will add the discussion in the revised manuscript.
>
> - **Modeling Human Behaviors is Challenging:** We agree that human discretion models may be hard to accurately predict, which is the main reason we proposed a UCB and TS variant to better explore the unknown human behaviors. In addition, we can use a non-parametric learner such as Gaussian Process (GP) or BNN (by universal approximation theorem) to learn the human discretion behavior that guarantees the true model can be learned with enough samples.
> Furthermore, recent studies and real-world applications have demonstrated strong predictive performance of human discretion behaviors [1,2] or human mental models [2,3], which indicates the human discretion model can be learned in practice, which we hope can address your concern.
>
> - **Changing Human Discretion Model:** We agree that human decision-making processes can change over time. We would like to kindly highlight that we discuss and evaluate this scenario in Appendix B where the true human behavior model changes at a given time step in the instance collection. Our method can still be applied in this setting and we find the Joint-BALD-UCB successfully selects the samples that are likely to be labeled and are informative to the predictive model and human discretion model, and it can better recover the correct decision boundary than baselines.
>
> - **Real-world application and Data sets:** Thanks for the suggestion. First, we kindly note that one of our experiments is motivated by a financial fraud investigation. Second, we added additional experimental results on Fashion MNIST, CIFAR-10, and two UCI datasets (Adult and Mushroom) in the rebuttal. The results are included in Figures 2 and 3 in the one-page PDF response. We find similar qualitative conclusions that Joint-BALD variants often produce better results than Naive-BALD and RANDOM across different costs. Though selective labeling is prevalent in industry use cases, it is challenging for us to find public real-world data with recorded human decision-maker judgments, so we choose to vary the human behavior model in the paper so that we can get a more detailed comparison across different methods.
>
> Please find the results for additional datasets and human behaviors in **Response to All**.
>
> - **Figures and Writings:** In the revised paper, we ensure all the figure legends are clear with proper fonts (like the figures in our PDF response). We also add intuitive explanations to the objective terms in the Definitions. We add implementation details of the training discretion model in the Appendix.
>
> - **Assumptions Biases:** We impose minimum assumption on the human discretion model since $e(x)$ can be any complex function. In practice, we can use a non-parametric learner such as Gaussian Process (GP) or BNN (by universal approximation theorem) to learn the human discretion behavior that guarantees the true model can be learned with enough samples.
>
> - **$c_e$ and $c_l$:** In the ALIR problem, we assume there is a cost to decide whether to label an instance (examine cost $c_e$) and a cost to actually label the instance (label cost $c_l$) (L130-131 in the main paper). The total cost is the sum of the current examination cost and label cost. We will make this more clear in the revised manuscript.
>
> We appreciate your insightful suggestions and have addressed it thoroughly in our revised manuscript to enhance the overall quality of the work. Please do let us know if you have any further concerns, or if this adequately addresses all the issues that you raised with the paper.
>
> [1] Sun, Jiankun, et al. "Predicting human discretion to adjust algorithmic prescription: A large-scale field experiment in warehouse operations." Management Science 68.2 (2022): 846-865.
>
> [2] Wang, Xinru, Zhuoran Lu, and Ming Yin. "Will you accept the ai recommendation? predicting human behavior in ai-assisted decision making." Proceedings of the ACM web conference 2022.
>
> [3] Bansal, Gagan, et al. "Beyond accuracy: The role of mental models in human-AI team performance." Proceedings of the AAAI conference on human computation and crowdsourcing. Vol. 7. 2019.
>
> [4] Madras, David, Toni Pitassi, and Richard Zemel. "Predict responsibly: improving fairness and accuracy by learning to defer." Advances in neural information processing systems 31 (2018).

---

> > ### Author Response · Authors · 2024-08-12
> >
> > Dear Reviewer fApQ,
> >
> > We hope this message finds you well. If you have any remaining questions or concerns, we would greatly appreciate your feedback. We believe we have thoroughly addressed the issues you raised and would be sincerely grateful if you could kindly reconsider your scores.
> >
> > Thank you,
> >
> > Authors of SEL-BALD

---

> > > ### Comment · Reviewer_fApQ · 2024-08-12
> > > **Rebuttal response**
> > >
> > > I thank the authors for their rebuttal. This has clarified many of my concerns. I still think that the robustness wrt a changing human discretion model deserves more attention in the main paper. Nevertheless, I have raised my score.

---

> > > > ### Author Response · Authors · 2024-08-12
> > > >
> > > > Thank you so much. We are very grateful to you for reading our rebuttal, and also increasing your score accordingly. We will incorporate all your inputs into the final version and make sure to include more discussions about changing human behaviors in Appendix B in the revised main paper.

---

### Official Review · Reviewer_WcLX · 2024-07-13

**Soundness:** 3
**Presentation:** 3
**Contribution:** 2
**Rating:** 4
**Confidence:** 4

**Summary:**

This paper considers an active learning scenario where the human annotation is done under the restriction that the annotator is biased in deciding the labeling. The Bayesian framework makes the e-BALD and various versions based on a posterior sample of labeling probabilities. The authors consider the three cases in human labeling probability, including the non-labeling of specific cases.  Three versions of the mean, quantile (UCB), and samples reveal different aspects in experiments. The trade-off between exploration and exploitation in labeling probability is observed. Using the quantile can have a detailed balance between examination and labeling (examination means checking the attribute and non-labeling).

**Strengths:**

The problem and task are novel. The BADL modified by the labeling probability is simple, clear, and well-defined. In experiments, the proposed algorithm performs better in the labeling rate for the candidates by the query.

**Weaknesses:**

There can be many issues.
The first one is the applicability of various metrics used in active learning.  Usually, entropy, margin, and variation ratio are traditional measures. Especially, the BALD combined with deep neural networks can show poor performance. It requires the study of applicability to other metrics or advanced active learning algorithms such as BADGE.
The second one concerns the datasets used in experiments. Three datasets are not sufficient; more datasets, such as Fashion MNIT or ImageNet-tiny, can be considered, and their statistical significance can be studied more thoroughly.
The last issue is human behavior, the scenario is relatively simple addressed by the simple logistic regression. The behavior of human annotators can be diverse because of their complex properties. It requires human models, which can be complex. When the human annotator's behaviors are complex, what’s the performance? Also, models such as logistic regression can be mimicked to generate samples.

Minor:
bayesian neural  -> Bayesian neural
e-bald -> e-BALD

**Questions:**

Q1: Can you provide other metrics, such as the F1 score? In my opinion, the 1-score can be lowered by selective labeling.
Q2: What’s the meaning of Total costs in the legend of Fig. 4?

**Limitations:**

Sufficiently discussed in the last section.

---

> ### Author Rebuttal · Authors · 2024-08-06
>
> We appreciate your recognition of the novelty of the problem and the clarity of our approach.
>
> **Other AL Metrics**: Thanks for the suggestion! While we mainly focus on the BALD objective in this paper, our methods can be generalized by replacing the mutual information with other metrics in the objective. For example, we can replace the information gain with other uncertainty measures such as entropy.
>
> As shown in Figure 1 in the response PDF, we implement Joint-Entropy-UCB with entropy measure on the synthetic data. Joint-Entropy-UCB can also improve the traditional entropy objective in this case. The BADGE method measures uncertainty at the gradient space. One potential way to apply BADGE is to compute the gradients for the last layer of the predictive model $p(y|x,\theta)$ and discretion model $p(a | x, \phi)$, weighted combine the gradients with the weights $(e(x), 1)$, and then use K-means++, however, this adaptation may be more challenging compared to traditional AL metrics and we believe will be interesting future work. We will discuss the connection with BADGE in the revised version.
>
> **Datasets and Human Behavior**: Thanks for the suggestion. We added additional experimental results on Fashion MNIST, CIFAR-10, and two UCI datasets (Adult and Mushroom). The results are included in Figures 2 and 3 in the PDF response. We find similar qualitative conclusions that Joint-BALD variants often produce better results than Naive-BALD and RANDOM across different costs.
>
> Following your suggestion, we also implemented a Logistic Regression based human behavior model in Figure 5 in the PDF response with similar qualitative conclusions as in the main paper.
>
> Please find the results for additional datasets, human behaviors, and more details in **Response to All**.
>
> **F1 Score:** In our experiments, we find F1 score often has a similar qualitative conclusion as the Accuracy. To illustrate that, we include a comparison between Accuracy and F1 score in Figure 4 in the PDF response for your reference. We will add the discussion about the F1 score in the paper.
>
> **Total Cost**: In the ALIR problem, we assume there is a cost to decide whether to label an instance (examine cost) and a cost to actually label the instance (label cost) (L130-131 in the main paper). The total cost is the sum of the current examination cost and label cost (the cost structure is defined in L212-213 in the paper). We will make this more clear in the revised manuscript.
>
> We appreciate your insightful suggestions and have addressed it thoroughly in our revised manuscript to enhance the overall quality of the work. Please do let us know if you have any further concerns, or if this adequately addresses all the issues that you raised with the paper.

---

> > ### Comment · Reviewer_WcLX · 2024-08-11
> > **Reply to Rebuttal**
> >
> > Thanks for your reply. The added results seem better during the rebuttal period. Also, the adaptation to other AL algorithms can be limited to the Bayesian framework, which is natural.  I have one more question about human behavior. The terminology of "Human behavior" is limited, in my opinion, since the rejection can occur by other machines. Can the case that some rejection or other actions can bother the proposed AL alg.'s performance exist?

---

> > > ### Author Response · Authors · 2024-08-11
> > >
> > > Thanks for the question. In our motivation, a human insurance agent needs to investigate a potential fraud case or a human doctor needs to perform a biopsy, which is why we describe it as human behavior. It may also be possible that a firm deploys another AI to determine the risk of such a labeling procedure. In this case, since the firm has query access, $e(x)$ is either known or can be estimated accurately with a large number of queries, then based on our theoretical analysis, the company can directly use $e$-BALD as the active learning strategy. The unknown human behavior brings additional challenges to this problem. We will add this discussion to the revised manuscript.

---

> > > > ### Author Response · Authors · 2024-08-12
> > > >
> > > > Dear Reviewer WcLX
> > > >
> > > > We sincerely hope that our response has adequately addressed the major points raised in your review. We would be very grateful if you could kindly let us know if you have any remaining reservations or if all of your concerns have been resolved. We would also greatly appreciate it if you could reconsider your scores in light of our clarifications.
> > > >
> > > > Thank you,
> > > >
> > > > Authors of SEL-BALD

---

> > > > > ### Comment · Reviewer_WcLX · 2024-08-12
> > > > > **Reply**
> > > > >
> > > > > Thanks for your reply. My remaining interest is when the $e(x)$ can bother the performance of AL to ensure the robustness of the proposed algorithm. Is it impossible by the theoretical validation?

---

> > > > > > ### Author Response · Authors · 2024-08-12
> > > > > > **question about $e(x)$**
> > > > > >
> > > > > > Thank you for your question. There are indeed scenarios where the function $e(x)$ may not be accurately learned, which could affect the performance of the proposed algorithm. One potential issue is the misspecification of the model class used to represent human behavior. For instance, if a linear model like Logistic Regression is applied to a fundamentally non-linear human behavior pattern, this could introduce estimation bias into the proposed AL method. To mitigate this, non-parametric models could be employed, as they are less prone to such misspecification bias.
> > > > > >
> > > > > > Another challenge involves measurement bias in the human behavior model. For example, the features recorded in the data may not fully capture the features humans actually perceive, leading to further inaccuracies in modeling human behavior. While addressing this issue is beyond the scope of our current work, future research could explore bounding the measurement error using various distance metrics [1] such as TV distance $TV(p(x),p(\tilde{x}))\leq\delta$ to define possible ranges for $e(x)$ and optimize over the worst case to ensure robustness.
> > > > > >
> > > > > > We will add the above discussion in the revised manuscript.
> > > > > >
> > > > > > [1] Guo, W., Yin, M., Wang, Y., & Jordan, M. (2022, June). Partial identification with noisy covariates: A robust optimization approach. In Conference on causal learning and reasoning (pp. 318-335). PMLR.

---

> > > > > > > ### Author Response · Authors · 2024-08-13
> > > > > > >
> > > > > > > Dear Reviewer WcLX,
> > > > > > >
> > > > > > > Could you let us know if our response addresses your question? Thank you.
> > > > > > >
> > > > > > > Best,
> > > > > > >
> > > > > > > Authors of SEL-BALD

---

### Author Rebuttal · Authors · 2024-08-06

# Response to All:

We thank the reviewers for your positive and constructive feedback. Your comments signiﬁcantly help us improve the paper.

## Datasets

In addition to the synthetic data, MNIST, and the loan fraud detection data in the paper, we further evaluate the proposed and the baseline methods on Fashion MNIST, CIFAR-10, and two UCI datasets (Adult and Mushroom). The results are included in Figures 2 and 3 in the one-page PDF response. We find similar qualitative conclusions that Joint-BALD variants often produce better results than Naive-BALD and RANDOM across different costs.
Please refer to the one-page PDF response for details.

## Human Behavior

We agree with the reviewers that a human annotator’s behavior can be complex. We would like to note that the proposed Joint-BALD does not pose functional restrictions on the human discretion model $e_\phi(x)$, which can be an arbitrarily complex function. For example, it can be modeled by a nonparametric learner like Gaussian Process or Bayesian neural networks.

Though selective labeling is prevalent in industry use cases, it is challenging for us to find public real-world data with recorded human decision-maker judgments, so we choose to vary the human behavior model in the paper so that we can get a more detailed comparison across different methods. In the paper, we examined homogeneous (Appendix C) and more complex heterogeneous human behaviors and found the advantages of Joint-BALD and its variants for unknown behavior heterogeneity. We also illustrate the advantage of Joint-BALD in adapting to changing human behavior (Appendix B). In addition, following reviewer WcLXs’ suggestion, we add an experiment using the Logistic Regression modeled human behavior model in Figure 5 in the one-page PDF response. More specifically, we randomly select 3 samples with the positive and negative class each and train a logistic regression to mimic the human decision behavior, then human labeling decisions are generated using the logistic regression. We find a similar conclusion that Joint-BALD variants often produce better results than Naive-BALD and RANDOM across different costs as demonstrated in the main paper.

---

### Comment · Area_Chair_ESN5 · 2024-08-11
**Author Rebuttal Available**

Dear Reviewers,

Thank you for your reviews and your time.  The authors have provided feedback to your questions as well as additional empirical results.  Please take a careful look and provide any follow up questions and update your reviews as necessary.

Thank you,
Area Chair for Submission 12433

---

### Public Comment · ~Niek_Tax1 · 2024-12-08
**Several related works that study the same problem setting were missed**

We have been in touch with authors by e-mail and made them aware that, although the problem setting is itself claimed as a novel contribution in the paper, identical problem settings have been studied in several earlier works [1,2,3,4,5].

The chosen solution direction of SEL-BALD remains novel, as none of those prior works have proposed solution directions based on extensions of the BALD method.

The authors have acknowledged by e-mail that these works study the same problem setting, and have committed to add discussions of those related work, and to modify the contribution statement to those prior works.

[1] Robinson, T. S., Tax, N., Mudd, R., & Guy, I. (2024). Active learning with biased non-response to label requests. Data Mining and Knowledge Discovery, 1-24.

[2] Fang, M., Zhu, X., & Zhang, C. (2012). Active learning from oracle with knowledge blind spot. In Proceedings of the AAAI Conference on Artificial Intelligence (Vol. 26, No. 1, pp. 2421-2422).

[3] Yan, S., Chaudhuri, K., & Javidi, T. (2015). Active learning from noisy and abstention feedback. In 2015 53rd Annual Allerton Conference on Communication, Control, and Computing (Allerton) (pp. 1352-1357). IEEE.

[4] Nguyen, C. V., Ho, L. S. T., Xu, H., Dinh, V., & Nguyen, B. T. (2022). Bayesian active learning with abstention feedbacks. Neurocomputing, 471, 242-250.

[5] Amin, K., DeSalvo, G., & Rostamizadeh, A. (2021). Learning with labeling induced abstentions. Advances in Neural Information Processing Systems, 34, 12576-12586.

---

### Decision · Program_Chairs · 2024-09-25

**Decision:**

Accept (poster)

**Comment:**

After discussion with the reviewers and a robust rebuttal from the authors (including additional results based on feedback), I recommend accepting the submission (poster).

The main concerns were around limited (1) a limited number of datasets (the authors have added 3 more), (2) simplistic human discretion models (the authors have added an ml-based human model), (3) lack of experiments with real human data (I tend to agree this is beyond the scope of typical academic settings).

I expect the authors to include the new results in the final version of the paper and expand on what could not be fit into the rebuttal pdf (e.g., include ML-based human model results for all datasets, include full set of experiments with uncertainty based sampling AL methods). Also, move some of the results and discussion of the dynamic/changing rater setting to the main paper.